SciPost Physics

Submission

# Optical lattice with spin-dependent sub-wavelength barriers

E. Gvozdiovas, P. Račkauskas, G. Juzeliūnas*

Institute of Theoretical Physics and Astronomy, Vilnius University, Saulėtekio 3, Vilnius LT-10257, Lithuania
* gediminas.juzeliunas@tfai.vu.lt

October 5, 2021

## Abstract

We analyze a tripod atom light coupling scheme characterized by two dark states playing the role of quasi-spin states. It is demonstrated that by properly configuring the coupling laser fields, one can create a lattice with spin-dependent sub-wavelength barriers. This allows to flexibly alter the atomic motion ranging from atomic dynamics in the effective brick-wall type lattice to free motion of atoms in one dark state and a tight binding lattice with a twice smaller periodicity for atoms in the other dark state. Between the two regimes, the spectrum undergoes significant changes controlled by the laser fields. The tripod lattice can be produced using current experimental techniques. The use of the tripod scheme to create a lattice of degenerate dark states opens new possibilities for spin ordering and symmetry breaking.

# 1  Introduction

Traditionally, optical lattices are created by interfering two or more light beams, so that atoms are trapped at minima or maxima of the emerging interference pattern depending on the sign of the atomic polarizability [1]. Optical lattices are highly tunable and play an essential role in manipulation of ultracold atoms [2–4]. The characteristic distances over which optical lattice potentials change are limited by diffraction and thus cannot be smaller than half of the optical wavelength $\lambda$. Yet the diffraction limit does not necessarily apply to optical lattices [5–9] or other sub-wavelength structures [10–17] relying on coherent coupling between atomic internal states.

It was demonstrated theoretically [5, 6, 8, 9, 18] and experimentally [19, 20] that a periodic array of sub-wavelength barriers can be formed for atoms populating a long lived dark state of the $\Lambda$-type atom-light coupling scheme. These barriers appear in regions of steep change in the dark state due to the geometric scalar (Born-Huang) potential [18, 21, 22]. The $\Lambda$ scheme has a single dark state, so no spin (or quasi-spin) degree of freedom is involved for the atomic motion in the dark state manifold affected by the sub-wavelength barriers.

Here we consider a way of creating a periodic array of spin-dependent sub-wavelength barriers that act differently on atoms depending on their internal state. For this we analyze a tripod atom light coupling scheme [23–26] characterized by two dark states playing the role of quasi-spin states. Previously, the tripod scheme was studied for creating a monopole field [24, 25, 27] and homogenous spin-orbit coupling [28–32], as well as for cooling atoms and ions [33, 34], but not for producing a lattice characterized by spin-dependent sub-wavelength barriers. Inclusion of the spinor degree of freedom provides new possibilities for controlling the spectral and kinetic properties of atoms in the lattice.

This paper is structured as follows. In the following Sec. 2 we introduce the tripod scheme of atom-light coupling characterized by two degenerate dark states, and assume the spatially periodic atom-light coupling. In Sec. 3 we consider the adiabatic motion of atoms in the manifold of two dark states playing the role of quasi-spin states. The atomic motion is then affected by a periodic array of sub-wavelength potential barriers acting mostly on the second dark state, as well as by a matrix-valued vector potential inducing transitions between the dark states. In Sec. 4 the spectrum of the tripod lattice is analyzed using the exact diagonalization of the Hamiltonian including all four atomic states, as well as exactly solving the eigenvalue problem for the adiabatic atomic motion projected onto the dark state manifold. The latter spectrum is also compared with the one relying on the scattering approach within the dark state manifold. The concluding Sec. 5 summarizes our findings and discusses the experimental implementation of the tripod lattice. Some details on calculating the Wannier functions are presented in Appendix A. Codes of the calculations and technical documentation detailing the numerical method are available in the Supplementary Material [35].

## 2 Tripod coupling scheme

### 2.1 Hamiltonian

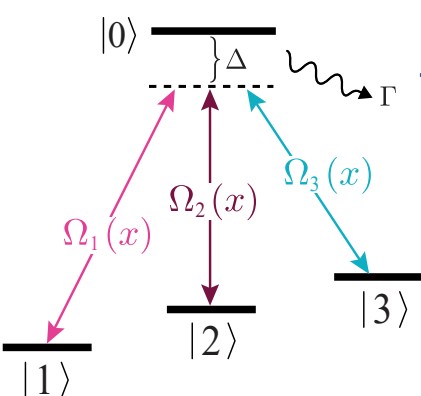

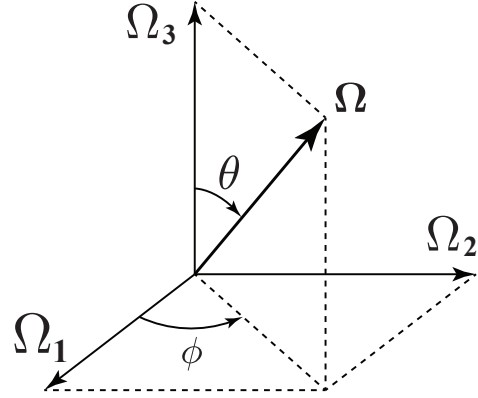

Figure 1: Tripod atom light coupling scheme.

Figure 2: Representation of a set of Rabi frequencies $\boldsymbol{\Omega} = (\Omega_1, \Omega_2, \Omega_3)$ in terms of the polar and azimuthal angles $\theta$ and $\phi$.

We consider atom-light interaction in the tripod scheme in which three laser beams characterized by the Rabi frequencies $\Omega_j = \Omega_j(x)$, with $j = 1, 2, 3$, couple three atomic ground states $|j\rangle$ to an excited state $|0\rangle$ detuned by $\Delta$ and characterized by the decay rate $\Gamma$, as illustrated in Fig. 1. The operator describing such coupling is given by in the rotating frame for the atomic internal states

$$\hat{V}(x) = \left(-\Delta - \frac{i}{2}\Gamma\right) |0\rangle\langle 0| + \sum_{j=1}^{3}\left[\frac{\Omega_j(x)}{2}|0\rangle\langle j| + H.c.\right], \tag{1}$$

where we have set $\hbar = 1$, so the energy is measured in units of frequency.

To create spin-dependent sub-wavelength barriers along the $x$ axis, we consider the following configuration of Rabi frequencies

$$\Omega_1(x) = \Omega_p, \tag{2}$$
$$\Omega_2(x) = \Omega_p \cos(2\pi x/a + \alpha), \tag{3}$$
$$\Omega_3(x) = \Omega_c \sin(2\pi x/a), \tag{4}$$

where $\alpha$ is the spatial phase difference between the standing waves $\Omega_2(x)$ and $\Omega_3(x)$, and $a$ is the lattice spacing. The amplitudes of the first two laser fields $\Omega_1(x)$ and $\Omega_2(x)$ (to be referred to as the probe fields) are taken to be much smaller than the amplitude of the control field $\Omega_3(x)$:

$$\epsilon = \Omega_p/\Omega_c \ll 1. \tag{5}$$

In that case the control field is dominant everywhere except around its zeros at $x = na/2$, with $n$ being an integer. This results in a periodic array of state-dependent sub-wavelength barriers at $x = na/2$, as will be shown in Sec. 3.2.2. At each barrier the ratio between the probe beams

$\Omega_2/\Omega_1 = (-1)^n \cos\alpha$ depends on the angle $\alpha$, so by changing $\alpha$ one can significantly alter the spin-dependence of the sub-wavelength lattice.

It is convenient to express the real valued Rabi frequencies $\Omega_j \equiv \Omega_j(x)$ entering the atom-light interaction operator in terms of spherical angles $0 \le \theta \le \pi$ and $0 \le \phi < 2\pi$, given by (see Fig. 2)

$$\cos\theta = \Omega_3/\Omega\,, \quad \tan\phi = \Omega_2/\Omega_1\,, \tag{6}$$

where $\Omega = \sqrt{\Omega_1^2 + \Omega_2^2 + \Omega_3^2}$ is the total Rabi frequency. The zeros of the control field correspond to $\theta = \frac{\pi}{2}$.

By including the kinetic energy along the $x$ direction, the full atomic Hamiltonian reads

$$\hat{H} = \frac{p_x^2}{2m} + \hat{V}(x) \quad \text{with} \quad p_x = -\mathrm{i}\partial_x\,, \tag{7}$$

where $m$ is the atomic mass. The corresponding state-vector

$$|\psi(x)\rangle = \sum_{j=0}^{3} |j\rangle\psi_j(x) \tag{8}$$

describes a combined atomic internal and center of mass motion, where $\psi_j(x)$ is a wave-function for the atomic center of mass motion in the $j$th internal state.

## 2.2 Spatial shift by half the lattice constant

The Hamiltonian (7) is invariant with respect to the spatial shift by the lattice constant $a$, such that $\left[\hat{H}, e^{\mathrm{i}p_x a}\right] = 0$. On the other hand, since the Rabi frequencies $\Omega_1(x)$ and $\Omega_{2,3}(x)$ given by Eqs. (2)-(4) are, respectively, symmetric and anti-symmetric with respect to the spatial shift by half the lattice constant $a/2$, the Hamiltonian also commutes with the combined shift operator $\hat{T}_{a/2} = \hat{U}e^{\mathrm{i}p_x a/2}$, which involves both a spatial $a/2$ shift and a change of atomic internal state, the latter described by the operator

$$\hat{U} = |2\rangle\langle 2| + |3\rangle\langle 3| - |1\rangle\langle 1| - |0\rangle\langle 0|\,, \tag{9}$$

with $\hat{U}^2 = \hat{I}$. Therefore the Hamiltonian and the translation operator $\hat{T}_{a/2}$ have a common set of eigenstates

$$\left|\psi^{(q)}(x)\right\rangle = \left|g^{(q)}(x)\right\rangle e^{\mathrm{i}qx}\,, \tag{10}$$

where $\left|g^{(q)}(x)\right\rangle$ is invariant with respect to the combined shift operator

$$\hat{T}_{a/2}\left|g^{(q)}(x)\right\rangle = \hat{U}\left|g^{(q)}(x+a/2)\right\rangle = \left|g^{(q)}(x)\right\rangle\,. \tag{11}$$

In that case the components of the expanded state-vector (8) read

$$\psi_j^{(q)}(x) = g_j^{(q)}(x)e^{\mathrm{i}qx}\,, \quad \text{with} \quad g_j^{(q)}(x+a/2) = \pm g_j^{(q)}(x)\,, \tag{12}$$

where the upper and lower signs correspond to $j = 2, 3$ and $j = 0, 1$, respectively. The solution given by Eqs. (10)-(12) is characterized by the quasi-momentum $q$ covering the first Brillouin zone (1BZ) $(-2\pi/a, 2\pi/a]$ which is twice larger than the 1BZ of the original Bloch solution for a lattice with periodicity $a$.

Neglecting the decay rate $\Gamma$, the system is governed by time-reversal symmetry, so the complex conjugation of the components $g_j^{(q)}(x)$, accompanied by reversion of the quasi-momentum $q \to -q$, does not change the eigenvalue equation giving $E_{-q} = E_q$ with $g_j^{(-q)}(x) = \left[g_j^{(q)}(x)\right]^*$. Such a condition is well maintained for atomic dynamics in the dark state manifold in which atomic decay is suppressed, as one can see in the energy dispersions presented in Figs 4 and 8 in Sec. 4.

# 3 Dark state representation

## 3.1 Dark states

The atom-light operator $\hat{V}(x)$ couples the excited state $|0\rangle$ to a special superposition of the ground states $|B\rangle \equiv |B(x)\rangle$, known as the bright state: $\hat{V}(x)|0\rangle = -\Omega|B\rangle$, with

$$|B\rangle = \sin\theta\left(|1\rangle\cos\phi + |2\rangle\sin\phi\right) + |3\rangle\cos\theta. \tag{13}$$

Two other superpositions of the ground states $|D_1\rangle \equiv |D_1(x)\rangle$ and $|D_2\rangle \equiv |D_2(x)\rangle$ are referred to as the dark states [23–25]. They are orthogonal between each other $\langle D_1|D_2\rangle = 0$ and to the bright state $\langle B|D_l\rangle = 0$, and thus are immune to the atom-light coupling, i.e. $\hat{V}(x)|D_l\rangle = 0$ with $l = 1, 2$.

The dark states are not uniquely defined. One of them, such as $|D_1\rangle$, can be chosen to be an arbitrary state vector orthogonal to $|B\rangle$. In that case the second dark state $|D_2\rangle$ should be orthogonal to both $|B\rangle$ and $|D_1\rangle$. A common choice for the dark states is the following [23–25]: the first state $|D_1\rangle$ is taken to be the dark state of the $\Lambda$ system composed of the bare states $|1\rangle$ and $|2\rangle$ :

$$|D_1\rangle = |1\rangle\sin\phi - |2\rangle\cos\phi, \tag{14}$$

so the second dark state is then given by

$$|D_2\rangle = \cos\theta\left(|1\rangle\cos\phi + |2\rangle\sin\phi\right) - |3\rangle\sin\theta. \tag{15}$$

The first dark state (14) is characterized by the angle $\phi$ and thus depends exclusively on the ratio between the Rabi frequencies of the first two laser beams $\Omega_1/\Omega_2$. The second dark state (15) additionally depends on the Rabi frequency of the third laser beam $\Omega_3$ through the angle $\theta$. As we will see in Sec.3.2, this provides steep potential barriers for atoms in the second dark state at the zero points of the dominant Rabi frequency $\Omega_3$, where the derivative $\theta'$ becomes much larger than the inverse lattice constant $1/a$.

## 3.2 Effective Hamiltonian for dark state dynamics

### 3.2.1 Projection onto the dark state manifold

If the total Rabi frequency $\Omega$ is large enough compared to the characteristic energy of the atomic center of mass motion, one can employ the adiabatic approximation by restricting the atomic motion to the manifold of degenerate dark states described by a projected state vector

$$|\psi_D(x)\rangle = I^D|\psi(x)\rangle = \sum_{l=1}^{2}|D_l\rangle\psi_{D_l}(x), \tag{16}$$

where $\psi_{D_l}(x)$ is the wave-function for the atomic center of mass motion in the $l$-th dark state, and $I^D = \sum_{l=1}^{2} |D_l\rangle \langle D_l|$ is the unit operator for projection onto the dark state manifold. Thus the atomic motion is described by a two-component wave-function

$$\psi_D(x) = \begin{pmatrix} \psi_{D_1}(x) \\ \psi_{D_2}(x) \end{pmatrix}. \tag{17}$$

The corresponding $2 \times 2$ Hamiltonian governing the evolution of such a spinor wave-function $\psi_D(x)$ reads

$$\mathcal{H}_D = \frac{1}{2m} \left(-i\partial_x - \mathcal{A}_D(x)\right)^2 + \mathcal{V}_D(x), \tag{18}$$

where the $2 \times 2$ matrices $\mathcal{A}_D(x)$ and $\mathcal{V}_D(x)$ are the geometric vector and scalar potentials presented in Ref. [25] for the general tripod atom-light coupling scheme. For the real valued Rabi frequencies $\Omega_j$ considered here, the geometric vector potential $\mathcal{A}_D(x)$ is proportional to the Pauli matrix $\sigma_y$:

$$\mathcal{A}_D(x) = \phi' \cos(\theta)\sigma_y, \quad \text{with} \quad \sigma_y = \begin{pmatrix} 0 & -i \\ i & 0 \end{pmatrix}. \tag{19}$$

On the other hand, the geometric scalar potential $\hat{\mathcal{V}}_D(x)$ can be expressed as a product of the row vector $\begin{pmatrix} c_1, & c_2 \end{pmatrix} = \begin{pmatrix} \phi' \sin\theta, & -\theta' \end{pmatrix}$ and its corresponding column vector:

$$\mathcal{V}_D(x) = \frac{1}{2m} \begin{pmatrix} c_1 \\ c_2 \end{pmatrix} \begin{pmatrix} c_1, & c_2 \end{pmatrix}. \tag{20}$$

Here the primes label the spatial derivatives of the angles $\theta$ and $\phi$ that parametrize the Rabi frequencies of the laser fields through Eq. (6).

For the Rabi frequencies given by Eqs. (2)–(4), both dark states are invariant with respect to the combined $a/2$ shift: $\hat{T}_{a/2}|D_l(x)\rangle = |D_l(x)\rangle$. Consequently the periodicity of the geometric vector and scalar potentials is half of the original lattice constant $a$:

$$\mathcal{A}_D(x + a/2) = \mathcal{A}_D(x), \quad \text{and} \quad \mathcal{V}_D(x + a/2) = \mathcal{V}_D(x),$$

so the eigenstates of the $2 \times 2$ matrix Hamiltonian $H_D$ are the two component Bloch spinors $\psi_D(x) \equiv \psi_D^{(q)}(x)$ with periodicity $a/2$:

$$\psi_D^{(q)}(x) = g_D^{(q)}(x) e^{iqx}, \quad \text{with} \quad g_D^{(q)}(x + a/2) = g_D^{(q)}(x). \tag{21}$$

### 3.2.2 Spin-dependent sub-wavelength barriers

The state-dependent scalar potential (20) affects the atoms in a special superposition $|\tilde{D}\rangle \equiv |\tilde{D}(x)\rangle$ of the original dark states $|D_1\rangle \equiv |D_1(x)\rangle$ and $|D_2\rangle \equiv |D_2(x)\rangle$ proportional to

$$|\tilde{D}\rangle \propto |D_1\rangle c_1 - |D_2\rangle c_2. \tag{22}$$

The coefficient $c_2 = -\theta'$ is associated with changes in the second dark state, and its absolute value is much larger than the inverse lattice constant $1/a$ at the zeros of the dominant Rabi frequency $\Omega_3(x)$ (appearing at $x = na/2$ with integer $n$), as one can see in Fig. 3. On the other hand, the angle $\phi$ is defined by the ratio of the first two Rabi frequencies $\Omega_2/\Omega_1$ and

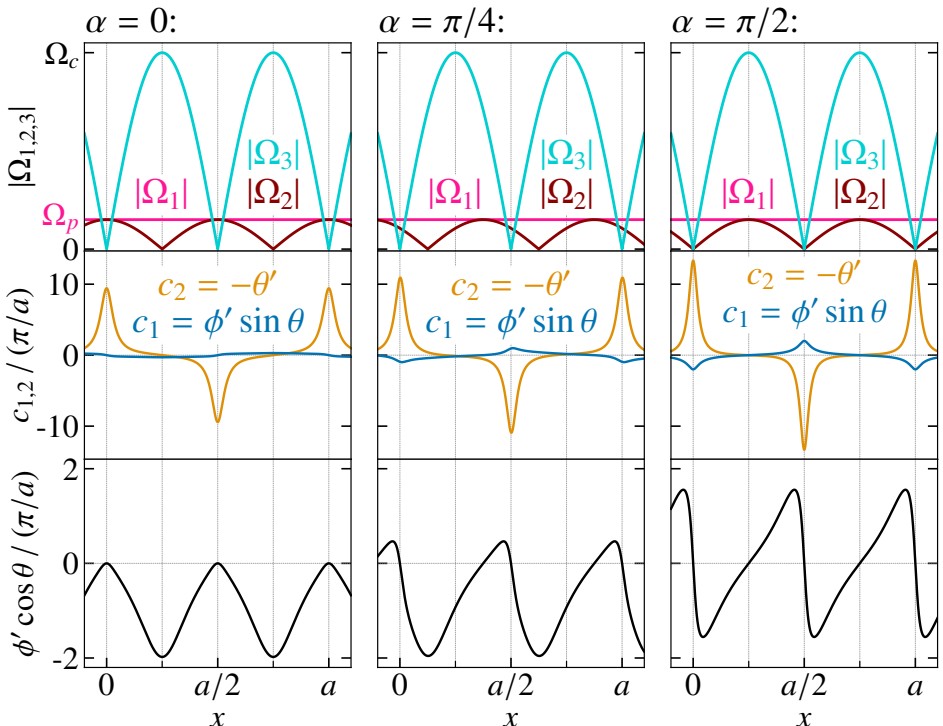

Figure 3: The modulus of the Rabi frequencies $\Omega_{1,2,3}(x)$ (first row), the quantities $c_1 = \phi' \sin\theta$ and $c_2 = -\theta'$ defining the geometric scalar potential $\mathcal{V}_D(x)$ in Eq.(20) (second row), as well as the quantity $\phi' \cos\theta$ entering the geometric vector potential $\mathcal{A}_D(x)$ in Eq. (19) (third row) for $\epsilon = \Omega_p/\Omega_c = 0.15$ and various values of $\alpha$.

thus changes smoothly, so the absolute value of the coefficient $c_1 = \phi' \sin\theta$ is of the order of $1/a$ or smaller. This means that at $x = na/2$, the factor $c_1$ associated with the first dark state plays a considerably less important role in the superposition (22) than the factor $c_2$ associated with the second dark state, as shown in Fig. 3. Furthermore, one can neglect $c_1$ also beyond the points where $x = na/2$. Using these arguments, one can simplify Eq. (20) by omitting $c_1$, leading to the following approximate scalar potential

$$\mathcal{V}_D(x) \approx \frac{(\theta')^2}{2m} \begin{pmatrix} 0 & 0 \\ 0 & 1 \end{pmatrix}. \tag{23}$$

Calling on the relation $\theta' = -(\cos\theta)'/\sin\theta$, together with Eq. (6) for $\cos\theta = \Omega_3/\Omega$ and Eqs.(2)-(4) for $\Omega_{1,2,3}$, one finds

$$\frac{(\theta')^2}{2m} = \frac{\pi^2 \epsilon^2}{2ma^2} \frac{[\cos(kx + 2\alpha) + 3\cos(kx)]^2}{[\cos^2(kx + \alpha) + 1][\epsilon^2 \cos^2(kx + \alpha) + \sin^2(kx) + \epsilon^2]^2}, \tag{24}$$

where $k = 2\pi/a$. Since $\epsilon = \Omega_p/\Omega_c \ll 1$, Eq. (24) represents a periodic set of barriers positioned at $x = na/2$, where neighboring barriers are separated by a distance of $a/2$, much larger than the characteristic width of each barrier. Thus one can represent Eq. (24) as a sum of narrow

potential barriers

$$\frac{(\theta')^2}{2m} \approx \sum_n v\left(x - na/2\right), \quad \text{with} \quad v\left(x\right) = \frac{k^2}{2m}\frac{\tilde{\epsilon}^2}{(\tilde{\epsilon}^2 + k^2 x^2)^2}, \tag{25}$$

where

$$\tilde{\epsilon} = \frac{1}{2}\epsilon\sqrt{\cos 2\alpha + 3} \tag{26}$$

is the effective ratio of the Rabi frequencies.

In this way, only atoms in the second dark state are scattered by the periodic array of sub-wavelength potential barriers located at $x = na/2$. The barriers appear due to rapid changes in the mixing angle $\theta$ featured in Eq. (15) for the second dark state. On the other hand, between the barriers the vector potential $\mathcal{A}_D(x) \propto \sigma_y$ describes transitions between the dark states (see Fig. 3). These two processes determine the shape of the energy dispersion. Approaching the barriers at $x = na/2$, the second dark state reduces to

$$|D_2\rangle = \frac{|1\rangle + (-1)^n |2\rangle \cos\alpha}{\sqrt{1 + \cos^2\alpha}}\cos\theta - |3\rangle\sin\theta \tag{27}$$

for $|x - na/2| \ll a$. Consequently, if $\cos\alpha \neq 0$, the dark state $|D_2\rangle \equiv |D_2(x)\rangle$ represents a different physical state in the vicinity of the barriers at $x = na/2$ for even and odd values of $n$, leading to scattering of different internal states at even and odd barriers.

Applying the methods presented in the Supplementary material of Ref. [5], the atom in the second dark state with the kinetic energy $E$ tunnels over an individual potential barrier $v(x)$ given by Eq. (25) with the following transition and reflection amplitudes $t = t(E)$ and $r = r(E)$ for $aQ\epsilon/\pi^2 \ll 1$:

$$t^{-1} = -1 + i\frac{\pi^2}{aQ\tilde{\epsilon}}, \quad \text{and} \quad r = -\left(1 - i\frac{aQ\tilde{\epsilon}}{\pi^2}\right)^{-1}. \tag{28}$$

where $Q = \sqrt{2mE}$ is the momentum corresponding to the energy $E$. We will use these amplitudes $t$ and $r$ to get an approximate dispersion represented by the $4 \times 4$ eigenvalue equation (30) for a periodic array of spin-dependent scatterers.

### 3.2.3 Dispersion for a periodic array of spin-dependent scatterers

To gain deeper insight into the tripod lattice, we present an effective approach based on scattering by individual barriers. Such an analysis relies on the fact that the distance between neighboring potential barriers positioned at $x = na/2$ is much larger than the characteristic width of each barrier $\zeta = \tilde{\epsilon}a/2\pi$. Further away from the potential barriers, one can neglect the potential $\mathcal{V}_D(x)$ in the Hamiltonian (18). In this spatial region, the atomic motion is affected only by the vector potential $\mathcal{A}_D = \phi'\cos\theta\sigma_y$, which induces transitions between the dark states due to the Pauli matrix $\sigma_y$. (The spatial dependence of $\phi'\cos\theta$ is plotted in Fig. 3 for different values of $\alpha$.) Therefore the Bloch eigensolution characterized by the quasi-momentum $q$ reads for $na/2 - \zeta < x < (n+1)a/2 + \zeta$, i.e. further away from the barriers:

$$\psi_D^{(q)}(x) = e^{i\gamma_n(x)\sigma_y}\left[B^{(q)}e^{iQ(x - na/2)} + C^{(q)}e^{-iQ(x - na/2)}\right]e^{iqla/2}, \tag{29}$$

with $\gamma_n(x) = \int_{na/2}^x d\tilde{x}\phi'_{\tilde{x}}\cos\theta$, where $\sigma_y$ is a Pauli matrix acting on the two component spinors $B^{(q)}$ and $C^{(q)}$.

Connecting the solution (29) at different sides of the barriers via the spin-dependent transition matrix $W_Q$ given by Eq. (31), one arrives at the eigenvalue equation

$$W_Q \begin{pmatrix} B_1^{(q)} \\ C_1^{(q)} \\ B_2^{(q)} \\ C_2^{(q)} \end{pmatrix} = e^{-iqa/2} \begin{pmatrix} B_1^{(q)} \\ C_1^{(q)} \\ B_2^{(q)} \\ C_2^{(q)} \end{pmatrix}. \tag{30}$$

The $4 \times 4$ matrix $W_Q$ has the following block structure

$$W_Q = e^{i\gamma_0 \sigma_y} \begin{pmatrix} I_Q & 0 \\ 0 & I_Q M \end{pmatrix} = \begin{pmatrix} I_Q \cos\gamma_0 & I_Q M \sin\gamma_0 \\ -I_Q \sin\gamma_0 & I_Q M \cos\gamma_0 \end{pmatrix}, \tag{31}$$

with

$$\gamma_0 = -\int_0^{a/2} d\tilde{x} \phi'_{\tilde{x}} \cos\theta, \tag{32}$$

where $I_Q = \mathrm{diag}\left(e^{-iQa/2}, e^{iQa/2}\right)$ is a diagonal $2 \times 2$ matrix reflecting the phases acquired during the forward and backward propagation of atoms between the barriers. Here also

$$M = \begin{pmatrix} \frac{1}{t(E)} & \frac{r^*(E)}{t^*(E)} \\ \frac{r(E)}{t(E)} & \frac{1}{t^*(E)} \end{pmatrix} \tag{33}$$

is a matrix for scattering of atoms in the second dark state by a single barrier, $t(E)$ and $r(E)$ being the corresponding transmission and reflection coefficients, given by Eq. (28). The atoms in the first dark state are not affected by the potential barriers, hence the free propagation matrix $I_Q$ that is featured in the first line of the block diagonal matrix $\mathrm{diag}\,(I_Q, I_Q M)$ in Eq. (31). Finally, the operator $e^{-i\gamma_0 \sigma_y}$ entering $W_Q$ describes a transition between the dark states during propagation between the barriers. The $4 \times 4$ eigenvalue equation (30) provides a relation between the quasi-momentum $q$ and the momentum $Q = Q_q$ defining the dispersion $E_q = Q_q^2/2m$.

In Eq. (32) for $\gamma_0$, the function $\cos\theta$ is close to unity everywhere in the integrand except for a narrow range of distances close to the boundaries at $x = 0$ and $x = a/2$, where the control field $\Omega_3$ is no longer dominant. Thus, in the zero order of approximation, $\cos\theta$ can be omitted in Eq. (32), giving $\gamma_0 \approx \phi(0) - \phi(a/2)$. In particular, for $\alpha = 0$, one has $\phi(0) = \pi/4$ and $\phi(a/2) = -\pi/4$, so that $\gamma_0 \approx \pi/2$. In that case the eigenvalue equation given by Eqs. (30)–(31) reduces to

$$-I_Q M \begin{pmatrix} B_2^{(q)} \\ C_2^{(q)} \end{pmatrix} = e^{-iqa/2} \begin{pmatrix} B_1^{(q)} \\ C_1^{(q)} \end{pmatrix}, \quad -I_Q \begin{pmatrix} B_1^{(q)} \\ C_1^{(q)} \end{pmatrix} = e^{-iqa/2} \begin{pmatrix} B_2^{(q)} \\ C_2^{(q)} \end{pmatrix}. \tag{34}$$

Thus one arrives at the following approximate eigenvalue equation

$$-I_Q^2 M \begin{pmatrix} B_2^{(q)} \\ C_2^{(q)} \end{pmatrix} = e^{-iqa} \begin{pmatrix} B_2^{(q)} \\ C_2^{(q)} \end{pmatrix}, \tag{35}$$

where the $2 \times 2$ matrix $-I_Q^2 M$ describes the scattering of atoms in the second dark state at even or odd barriers, separated by the distance $a$ and positioned at $na$ or $(n + 1/2)\, a$. Hence

the scattering of atoms in the second dark state takes place at every second barrier. This is because propagation of atoms between neighboring barriers separated by $a/2$ is associated with the matrix $e^{i\gamma_0\sigma_y}$, converting the second atomic dark state into the first one, which is not affected by the barrier.

Equation (35) has the same form as the corresponding scattering equation for the $\Lambda$ subwavelength lattice [5], but with a twice larger periodicity. Furthermore, in comparison to the $\Lambda$-type coupling scheme, the quasi-momentum $q$ is shifted by $\pi/a$ due to the minus sign in the scattering matrix $-I_Q^2 M = -I_{2Q}M$, the sign change appearing due to the interchange between the dark states during propagation between the barriers. Thus in contrast to the $\Lambda$ setup [5, 6], the quasi-momentum $q = 0$ does not correspond to the energy minimum in the lowest energy band, as one can see in the exactly calculated energy dispersion for $\alpha = 0$ in Fig. 4(c).

Since we are considering the expanded 1BZ covering the range $-2\pi/a < q \le 2\pi/a$, the dispersion described by Eq. (35) is twice degenerate, the eigenenergies being the same for $q = q'$ and $q = q' + 2\pi/a$. In a more accurate analysis, one should take into account that $\gamma_0 \ne \pi/2$. This means that the energy dispersion no longer repeats itself by shifting the quasi-momentum $q \to q + 2\pi/a$ in the expanded 1BZ, which is apparent in the exactly calculated dispersion in Figs. 4(c)-(d).

In the limit where $\alpha \to \pi/2$, the angle $\gamma_0$ defined by the integral in Eq. (32) goes to zero, as in that case the vector potential shown in Fig. 3 averages to zero after integration, giving $\gamma_0 = 0$. As such, for $\alpha = \pi/2$, the eigenvalue equation given by Eqs. (30)–(31) shows that atoms move freely in the first dark state, while atoms in the second dark state are scattered at every barrier (in contrast to scattering at every second barrier for $\alpha = 0$). This is confirmed by the exact calculations of the spectrum in this limit presented in Sec. 4.3.

# 4   Analysis of the spectrum

The spectrum of the tripod lattice is analyzed using the exact diagonalization of the Hamiltonian including all four atomic states using numerical algorithms of Ref. [36–40], as well as exactly solving the eigenvalue problem for the adiabatic atomic motion projected onto the dark state manifold. The latter spectrum is also compared with solutions of the $4 \times 4$ eigenvalue equation (30) relying on the scattering approach within the dark state manifold.

## 4.1   Energy dispersion for small and moderate phase $\alpha$

Parts (a) and (b) of Fig. 4 display exact calculations of the three lowest dispersion branches for small values of the angle $\alpha$ characterizing the phase of the Rabi frequency of the second probe beam $\Omega_2(x)$ with respect to that of the control beam $\Omega_3(x)$ in Eq. (3). Black lines correspond to the real part $E'_{q,s}$ of the complex eigenvalues $E_q \equiv E_{q,s} = E'_{q,s} + iE''_{q,s}$, and red areas indicate the widths of the dispersion lines represented by the imaginary part $E''_{q,s}$, where the index $s = 1, 2, \dots$ labels the dispersion bands. The minima of the dispersion branches $E'_{q,s}$ are seen to fit well with the discrete spectrum of a square well with a width equal to $a$, which is twice larger than the spacing $a/2$ between the neighboring potential barriers:

$$E_s^{\min} = s^2 E_R \,, \tag{36}$$

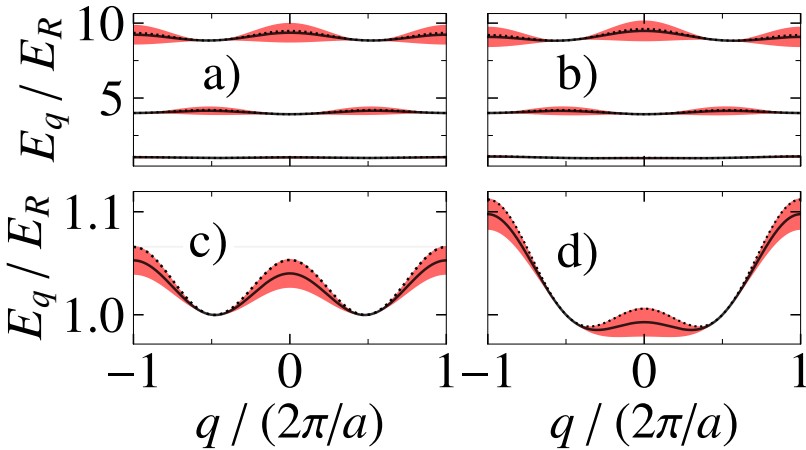

Figure 4: The first three Bloch bands with $s = 1, 2, 3$ [parts a),b)] and the first Bloch band with $s = 1$ [parts c),d)] in the expanded 1BZ. The plots in parts a),c) and and b),d) are made for $\alpha = 0$ and $\alpha = 15°$, respectively, with $\epsilon = \Omega_p/\Omega_c = 0.1$, $\Omega_p = 2000\, E_R$, $\Gamma = 1000\, E_R$ and $\Delta = \Omega_p$. In each band $s = 1, 2, 3$, solid lines correspond to the real part $E'_{q,s}$ of the complex eigenvalues $E_{q,s} = E'_{q,s} + iE''_{q,s}$, while red areas indicate the magnified widths of the dispersion lines represented by a),b) $40E''_{q,s}$ and c),d) $8E''_{q,s}$. Dotted lines show the energy dispersions obtained from the adiabatic Hamiltonian (18).

where $E_R = (\pi/a)^2/(2m)$ is the recoil energy corresponding to the momentum $\pi/a$. This confirms the conclusion reached in Sec. 3.2.3 that strong scattering of atoms takes place at every second barrier separated by $a$ for small values of phase $\alpha$.

The dispersions shown in Fig. 4 corresponding to rather small values of the angle $\alpha$ (0 and 15°) look quite similar on the large scale in parts (a)-(b) [1]. Figures 4(c)-(d) display the enlarged dispersion of the first energy band $s = 1$, showing a characteristic double well shape with a smaller height at the center $q = 0$ than at the edges $qa = \pm 2\pi$ of the expanded 1BZ. The linewidths are much smaller than the width of the energy bands. Losses are negligible in the lower part of the dispersion curves $E'_{q,s}$, corresponding to the minima of the double well dispersion at $qa \approx \pm\pi$ for odd bands $s = 1, 3, \ldots$, as well as at the center or edges of the expanded 1BZ at $qa = 0, \pm 2\pi$ for even bands $s = 2, 4, \ldots$. As discussed in Sec. 3.2.3, this is opposite to the $\Lambda$ scheme in which both the real part of the dispersion and losses are minimum at $q = 0$ for odd bands [5]. The dispersion shown in Fig. 4 can be well described in terms of the dissipative tight binding model including the nearest neighbor (NN) and the next nearest neighbor (NNN) couplings:

$$E_q = J_0 + J_1 \cos(qa/2) + J_2 \cos(qa) , \tag{37}$$

where $J_n = J'_n + iJ''_n$ with $|J'_n| \gg |J''_n|$. Interestingly, the NNN coupling is much more significant than the NN coupling, $|J'_2| \gg |J'_1|$ and $|J''_2| \gg |J''_1|$ for small $\alpha$. Furthermore, $J'_1$ and $J'_2$ have opposite signs, as one can see in Figs. 9 and 10 presented in Sec. 4.3. In particular, for the lowest band (s=1) one has $J'_1 < 0$ and $J'_2 > 0$, which explains the double well type dispersion featured in Fig. 4(c)-(d). Note also that the real and imaginary parts of the NNN matrix

---

[1]Changes in the dispersion curves become more visible on the large scale for bigger values of $\alpha$ shown in Fig. 5.

element $J_2'$ and $J_2''$ have opposite signs, with $J_2'$ being negative (positive) for even (odd) bands $s$. Therefore for even bands losses vanish at the dispersion minimum at $q = 0$, whereas losses for odd bands are negligible in the vicinity of the minima of the double well dispersion at $qa \approx \pm\pi$.

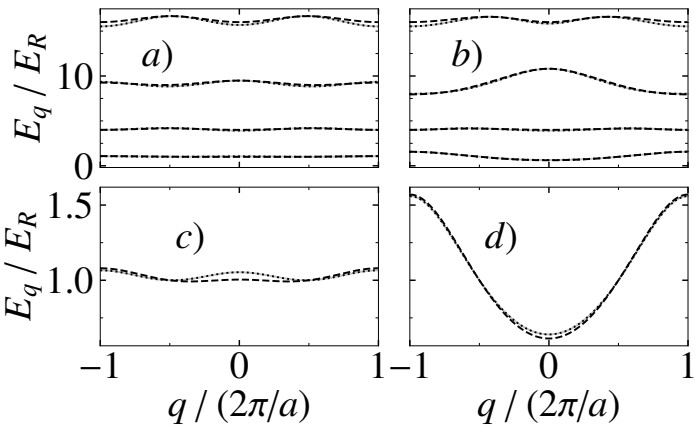

Figure 5: Top row: first 4 energy bands, bottom row: first energy band for $\Delta = 0$, $\epsilon = \Omega_p/\Omega_c = 0.1$, a),c) $\alpha = 0$ and b),d) $\alpha = \pi/4$. Gray lines are obtained by diagonalizing the full Hamiltonian (7), dotted lines in the same way from the adiabatic Hamiltonian (18), and dashed lines are evaluated from the scattering problem (34).

The solid dispersion lines in Figs. 4 and 5 agree well with the dispersions obtained for the adiabatic motion within the dark state manifold (dotted lines). More generally, we have checked that the effective dark state Hamiltonian (18) recreates the exact band structure for any $\alpha$, not necessarily small, as shown in Fig. 8 of Sec. 4.3, provided the detuning $\Delta$ is considerably smaller than the amplitude of the control field $\Omega_c$, and the ratio between the amplitudes of the probe and control fields is not too small, $\epsilon = \Omega_p/\Omega_c \gtrsim 0.05$. For smaller values of $\epsilon$, non-adiabatic losses become important due to significant changes in the composition of the atomic dark states at the potential barriers. Note that in Fig. 4, deviations of the adiabatic dispersion curves (dashed lines) from the exact dispersion (solid lines) are mainly due to non-zero values of $\Delta$ and $\Gamma$, the difference becoming minimum for zero detuning $\Delta = 0$ shown in Fig. 5.

The effective $4 \times 4$ eigenvalue equation (30) relying on the scattering approach within the dark state manifold reproduces the band structure of the odd energy bands accurately for $\alpha \gtrsim \pi/4$, $\Delta \ll \Omega_c$ and $\epsilon \gtrsim 0.05$, as one can see in parts b) and d) of Fig. 5. While the overall band structure and band gaps shown in parts a) and b) of Fig. 5 are maintained for any $\alpha$, the fine details of the even band energy dispersion curves are not precisely reproduced. Additionally, we note that the solutions of the scattering approach become less accurate for the higher energy bands.

## 4.2   Wannier functions

To gain more insight into the tight binding model, we have considered the multi-component Wannier function localized around $x = na/2$, with $n$ being an integer:

$$W_n(x) = [W_{n,D_1}(x), W_{n,D_2}(x), W_{n,B}(x), W_{n,0}(x)]^\dagger, \tag{38}$$

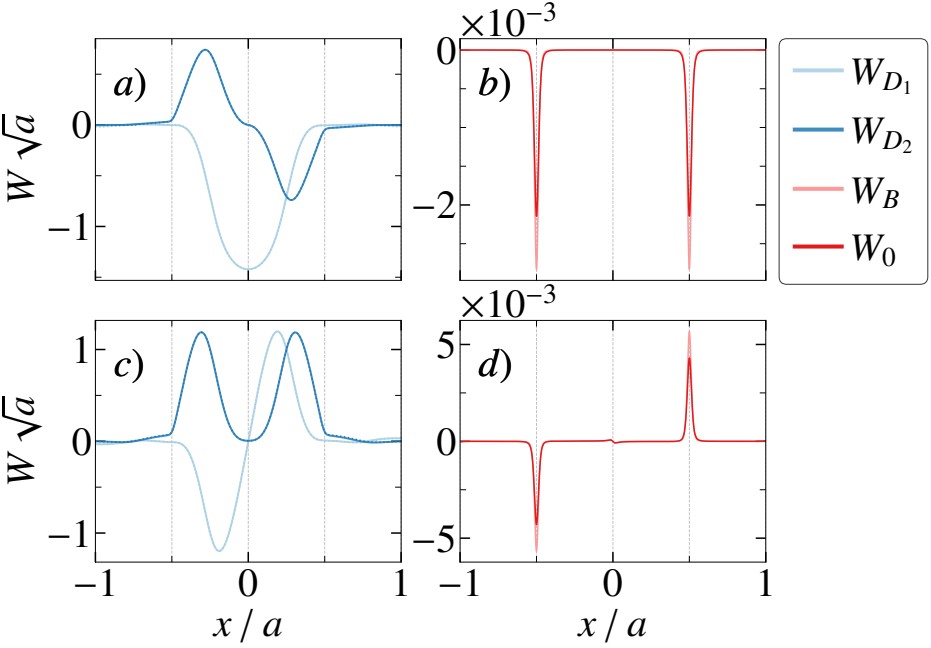

Figure 6: Exactly calculated Wannier functions centered at $x = 0$ for the dark states $W_{D_{1,2}}(x)$ in parts a) and c), as well as the bright state $W_B(x)$ and the excited state $W_0(x)$ in parts b) and d). The Wannier functions describe the first energy band in a),b) and the second band in c),d) in the expanded 1BZ. Parameters are taken to be $\epsilon = \Omega_p/\Omega_c = 0.1$, $\Delta = \Omega_p = 8000\,E_R$, $\Gamma = 4000\,E_R$ and $\alpha = 0$. Dotted lines in parts a) and c) are obtained by diagonalizing the effective dark state Hamiltonian (18). It is hard to see these lines since they coincide almost perfectly with the exact Wannier functions for the dark states.

where the components on the right-hand side $W_{n,Y}(x) = \langle Y(x)|W_n(x)\rangle$ with $Y = D_1, D_2, B, 0$ represent projections of the full Wannier state vector $|W_n(x)\rangle$ onto individual internal states. The constituting Wannier functions correspond to the

- dark states $W_{n,D_{1,2}}(x) = W_{D_{1,2}}(x - na/2)$,

- bright state $W_{n,B}(x) = (-1)^n W_B(x - na/2)$,

- excited state $W_{n,0}(x) = (-1)^n W_0(x - na/2)$,

where the alternating factor $(-1)^n$ reflects the fact that the bright and excited states alter their sign after applying a combined $a/2$ shift operator $\hat{T}_{a/2}$, defined in Sec. 2.2:

$$\hat{T}_{a/2}|B(x)\rangle = -|B(x)\rangle \quad \text{and} \quad \hat{T}_{a/2}|0\rangle = -|0\rangle. \tag{39}$$

As discussed in more detail in Appendix A, the Wannier state vector $|W_n(x)\rangle$ maximally localized around $x = na/2$ is obtained by superimposing the Bloch states of an individual band in the extended Brillouin zone with the phase factors $e^{-iqna/2}$. Thus the Wannier functions are strictly orthogonal for different values of $n$, despite the Hamiltonian being non-Hermitian due to the imaginary part in the excited state energy. The multicomponent Wannier functions

$W_n(x) \equiv W_{s,n}(x)$ with the same index $n$ but belonging to different Bloch bands $s$ are orthogonal to great accuracy, as we are considering the atomic motion in the dark state manifold with only a tiny contribution from the lossy excited state.

Figure 6 shows the Wannier functions $W_{D_{1,2}}(x)$, $W_B(x)$ and $W_0(x)$ centered at $x = 0$ for the first and second dispersion bands ($s = 1, 2$) and $\alpha = 0$. Despite the reduction of the unit cells to $a/2$ due to expansion of the 1BZ, the Wannier functions extend over a distance equal to the lattice constant $a$, as the atoms are confined between the barriers located at $x = \pm a/2$. These barriers scatter atoms in the second dark state $|D_2(x)\rangle$, so the Wannier function $W_{D_2}(x)$ dominates over $W_{D_1}(x)$ in the vicinity of the barriers. Between the barriers there is an exchange among the two dark states due to the vector potential term $\mathcal{A}_D(x) \propto \sigma_y$. The second dark state is thus converted to the first one at the center $x = 0$, where the Wannier function $W_{D_2}(x)$ vanishes, whereas $W_{D_1}(x)$ reaches its maximum magnitude for the first band, as shown in Fig. 6(a). Since only the second dark state is affected by the potential barriers, the barrier at $x = 0$ has little influence over the multi-component Wannier function centered at $x = 0$. The Wannier functions of the bright and excited states ($W_B(x)$ and $W_0(x)$) emerge due to non-adiabatic transitions. Thus they are localized close to the barriers at $x = \pm a/2$, where the Rabi frequency of the control field goes to zero (see Figs. 6(b),(d)). A tiny residual non-adiabatic effect due to the barrier at the center of the Wannier function appears as a small bump at $x = 0$ in Fig. 6(d) for the Wannier functions of the bright and excited states.

Atoms populating the multi-component Wannier function $W_n(x)$ centered at $x = na/2$ can tunnel to the next neighboring Wannier functions $W_{n\pm2}(x)$ centered at $x = (n \pm 2)a/2$ through the barriers located at $x = (n \pm 1)a/2$. These processes are described in terms of the matrix elements $J_2$ of the Hamiltonian between such next neighboring multi-component Wannier functions. A small imaginary part $J_2''$ emerges due to a small non-adiabatic contribution by the lossy excited state shown in Figs. 6(b),(d). On the other hand, the real part $J_2'$ is mostly due to the Wannier function of the second dark state $W_{D_2}(x)$, which is anti-symmetric with respect to the center $x = 0$ for the first band (see Fig. 6(a)), leading to the dispersion minima at non-zero quasi-momentum $q$ in the lowest band shown in Fig. 4(c)-(d).

## 4.3  Brick-wall lattice

The system can be visualized in terms of a two layer brick-wall lattice shown in Fig. 7 (a), in which the bricks in the lower (upper) layer represent the multi-component Wannier function $W_n(x)$ with odd (even) values of $n$. NNN coupling between the Wannier functions described by the matrix element $J_2$ corresponds to horizontal tunneling between the adjacent bricks of individual layers. Additionally, there is an inter-layer tunneling described by the NN coupling element $J_1$, which appears due to a residual effect of the barrier at the center of the Wannier function. For the odd bands, the effect of NN coupling becomes more important as $\alpha$ grows, leading to a more shallow double well dispersion in Fig. 4(d) in comparison to Fig. 4(c). In fact, for larger values of $\alpha$, the vector potential term converts the second dark state into the first one to a lesser extent at the center of the Wannier function, which increases scattering of atoms at $x = na/2$, and thus NN coupling becomes more significant.

By further increasing the phase $\alpha$, the band structure undergoes significant modifications, as illustrated in Fig. 8. For the first band, NN tunneling grows quickly with $\alpha$ and eventually overcomes NNN tunneling. The odd bands ($s = 1, 3, \ldots$) are very sensitive to $\alpha$ and eventually merge to form a free particle energy dispersion branch as $\alpha \to \pi/2$. In contrast, the even bands ($s = 2, 4, \ldots$) remain relatively intact with increasing $\alpha$; as $\alpha$ approaches $\pi/2$, they transform

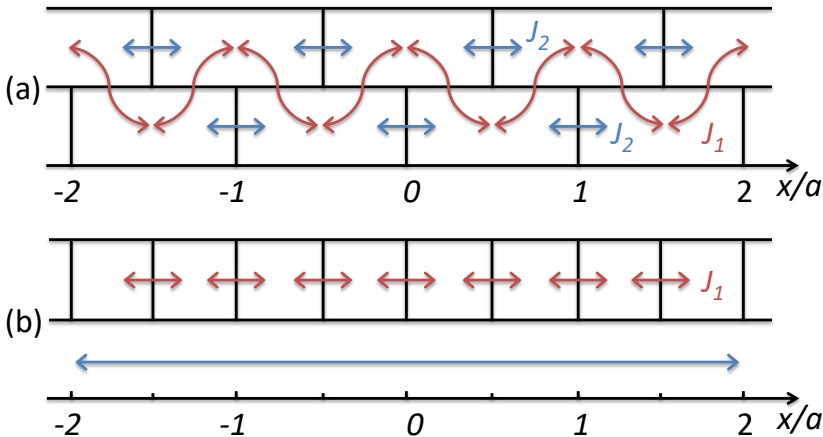

Figure 7: (a) Semi-synthetic lattice of the brick-wall-type for small $\alpha$. Tunneling between the bricks in each layer is described by the next nearest neighbor (NNN) coupling matrix element $J_2$. The layers are coupled by the nearest neighbor (NN) matrix element $J_1$. (b) For $\alpha \to \pi/2$ tunneling to the left or right occurs through barriers situated at $na/2$ for atoms in the second dark state (upper part), whereas atoms in the first dark state move freely (lower part).

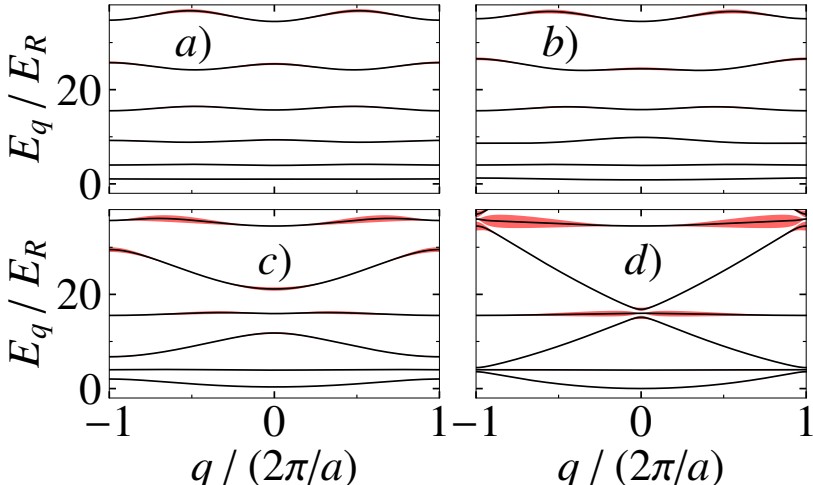

Figure 8: The first six Bloch bands ($s = 1, 2, \ldots 6$), obtained by diagonalizing the full Hamiltonian (7) in the expanded 1BZ for a) $\alpha = 0$, b) $\alpha = 30°$, c) $\alpha = 60°$ and d) $\alpha = 85°$ with $\epsilon = \Omega_p/\Omega_c = 0.1$, $\Omega_p = 2000\,E_R$, $\Gamma = 1000\,E_R$ and $\Delta = \Omega_p$. In all the plots the spectral line widths are magnified by a factor of 5.

into energy bands of $\Lambda$-like atom-light coupling [5,6], characterized by a twice smaller distance $a/2$ between the barriers. This can be understood by observing that, for $\alpha = 0$, the Wannier functions of both dark states vanish at the center $x = na/2$ for even bands (see Fig. 6(c)), so these Wannier functions are not as sensitive to the increasing effect of the barrier at $x = na/2$ as $\alpha$ grows. In the limit where $\alpha \to \pi/2$, tunneling to the left or right takes place at every barrier located at $na/2$ for atoms in the second dark state, as shown in the upper part of Fig. 7(b), whereas atoms move freely in the first dark state (lower part). This is because for

$\alpha = \pi/2$ there is no longer a vector potential type term converting the first dark state to the second one (and vice versa) during atomic propagation between the barriers. Note that the localization centers of the Wannier functions become shifted by $a/4$ in the upper part of Fig. 7(b) as a consequence of scattering at every barrier.

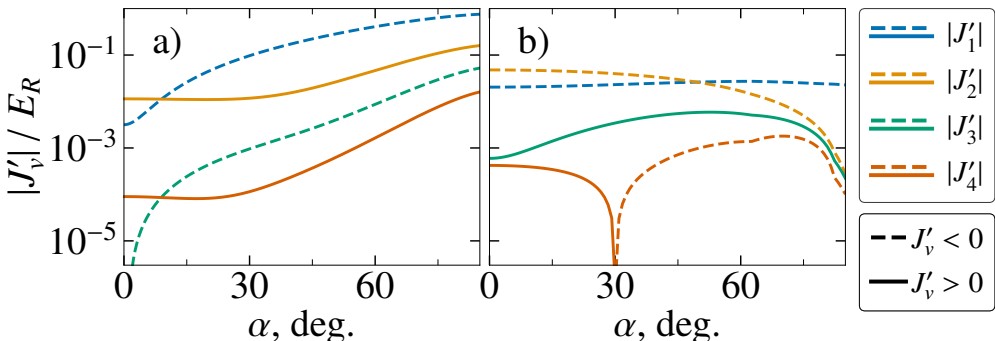

Figure 9: Absolute values of the first four tight binding tunneling parameters $J'_v$ for a) the first and b) second energy bands as a function of the phase $\alpha$ for $\epsilon = \Omega_p/\Omega_c = 0.1$ and $\Omega_p = 2000\,E_R$. Dashed (full) lines correspond to negative (positive) values of $J'_v$.

Figure 9(a) shows the $\alpha$ dependence of the tight binding tunneling parameters $J'_v$ between the first four neighboring lattice sites (v=1,2,3,4) for the first energy band ($s = 1$). As $\alpha$ becomes non-zero, one can see a steep increase in the tunneling matrix element $J'_v$ for odd values of $v$, whereas for even $v$ the parameter $J'_v$ start increasing at much larger values of $\alpha$. When $\alpha$ approaches $\pi/2$, all the tunnelling parameters $J'_v$ are a few orders of magnitude larger than in the case where $\alpha = 0$, as the atoms loaded into the bands with odd $s$ behave like free particles for $\alpha \to \pi/2$. We also note that the $4 \times 4$ eigenvalue equation (30) relying on the scattering approach accurately predicts the free particle behavior of bands with odd $s$ for $\alpha \to \pi/2$. For the second energy band (s=2) shown in Fig. 9(b), the phase $\alpha$ has a very different effect on the tunnelling parameters. As $\alpha$ approached $\pi/2$, all the tunnelling parameters $J'_v$ vanish, with the exception of the NN tunnelling corresponding to $v = 1$. As such, atoms loaded into the even bands are effectively confined by steep potential barriers at $x = na/2$ separated by $a/2$.

Up to now we have investigated the situation where the detuning $\Delta$ is considerably smaller than the amplitude of the control field $\Omega_c$. Going beyond this assumption, in Fig. 10 we observe that the NN tunneling matrix element $J'_1$ does not change with $\Delta$. On the other hand, the NNN tunneling element $J'_2$ decreases with $\Delta > 0$ and undergoes sign reversal after reaching the zero point, in which NNN tunneling is suppressed due to destructive interference between different components of the multi-component Wannier functions. When the NNN tunneling element $J'_2$ vanishes, only the small NN tunneling parameter $J'_1$ contributes to the energy dispersion, which results in an almost flat dispersion band. A similar effect can be observed also for the sub-wavelength lattice obtained using the $\Lambda$ coupling scheme, where the NN tunnelling element also goes to zero at a certain value of the detuning. For the tripod scheme, the detuning causes the NNN tunnelling element to vanish in the same fashion, as illustrated in Fig. 10, while the NN element is largely unaffected because the Wannier function is spread over two unit cells. Additionally, the tight binding parameters can be controlled with the Rabi frequency ratio $\epsilon = \Omega_p/\Omega_c$, as in the $\Lambda$ sub-wavelength lattice [19].

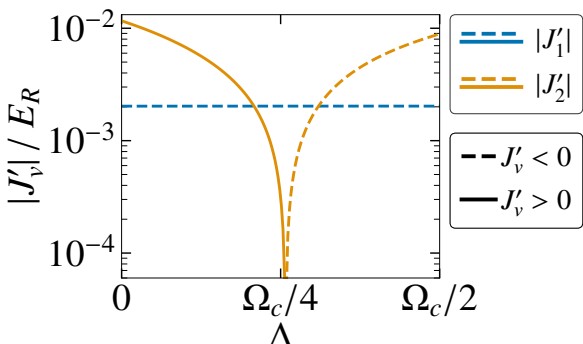

Figure 10: Absolute value of the first two tight binding tunneling parameters $J_1'$ and $J_2'$ for the first Bloch band as a function of detuning $\Delta$ for $\epsilon = \Omega_p/\Omega_c = 0.08$, $\Omega_p = 2000\,E_R$ and $\alpha = 0$. Dashed (full) lines correspond to negative (positive) values of $J_v'$.

## 5 Concluding remarks

We have analyzed the tripod atom light coupling scheme characterized by two dark states playing the role of quasi-spin states. It is demonstrated that by properly choosing the laser fields which couple the atomic internal states, one can create a lattice with spin-dependent sub-wavelength barriers acting differently on atoms in different atomic dark states. Inclusion of the spinor degree of freedom allows to flexibly alter the atomic motion ranging from the tight-binding type atomic dynamics in the effective brick-wall type lattice shown in Fig. 7(a) to the free motion of atoms in one dark state and the tight binding lattice with a twice smaller periodicity for atoms in another internal state, Fig. 7 (b). Between the two regimes, the spectrum undergoes significant changes controlled by the laser fields, as illustrated in Fig. 8.

The tripod scheme has already been experimentally realized by coupling three metastable atomic ground states to a common excited state [41, 42]. In particular, it was shown that the tripod setup can provide a robust and variable beam splitter for a beam of metastable neon atoms [41]. On the other hand, the tripod scheme was used to implement non-Abelian adiabatic geometric transformations for cold fermionic gas of strontium-87 atoms by laser coupling of three hyperfine ground states $|F_g = 9/2; m_g\rangle$ with $m_g = 9/2, 7/2, 5/2$ to their common excited state $|e\rangle = |F_e = 9/2; m_e = 7/2\rangle$ using two co-propagating beams with opposite circular polarizations and a third linearly polarized beam orthogonal to the first two beams [42]. The tripod-type scheme has also been implemented for photonic systems [43]. The lattice with spin-dependent sub-wavelength barriers considered here can be implemented using a setup similar to that in Ref. [42], with the co-propagating circularly polarized waves replaced by the standing waves describing the Rabi frequencies $\Omega_2(x)$ and $\Omega_3(x)$ in Eqs. (3)-(4). In this way, it is feasible to produce the sub-wavelength tripod lattice using currently available experimental techniques.

The lattice with sub-wavelength barriers has already been implemented for the $\Lambda$ scheme by adiabatically loading the ultracold atoms to the lowest band [19]. A similar adiabatic loading can be applied for the lattice with spin-dependent sub-wavelength barriers involving the tripod scheme. The atoms loaded to the lowest Bloch band will be maintained in it, as the difference in energies between the ground and first excited states is of the order of the recoil energy corresponding to temperature which greatly exceeds the nK temperature range

of the ultracold atomic gas. Since the tripod scheme has two degenerate dark states, there is an extra possibility to populate specific dark states during the adiabatic loading. This can be done by transferring the ultracold atoms to a desired dark state by the tripod STIRAP (stimulated adiabatic Raman passage) [23, 24] prior to the adiabatic loading.

The use of the tripod scheme to create a lattice of degenerate dark states opens new possibilities for spin ordering and symmetry breaking. In conventional spinor systems degeneracy is usually controlled by an external magnetic field that typically has noise on the scale of the spin-dependent interactions set either by exchange interactions or the native spin-dependent collisions. One might speculate that the present system would support new forms of magnetic ordering, or new routes to create established forms of magnetic order.

*Note added.* Shortly after submitting this manuscript to arXiv and SciPost, a related preprint appeared in arXiv [44] which also analyzes the spin-dependent sub-wavelength barriers using the tripod atom-light coupling scheme. The latter work complements the present study by covering some other aspects of the problem, including the case where the amplitudes of the probe fields $\Omega_{1,2}(x)$ are not the same.

## Acknowledgements

Helpful discussions with Egidijus Anisimovas, Tomas Andrijauskas, Julius Ruseckas and Ian B. Spielman are gratefully acknowledged.

**Author contributions**   G. J. conceived the idea which was subsequently refined together with E. G. and P. R. All authors participated in carrying out analytical investigations. E. G. created the computational codes and performed the numerical calculations. Some additional numerical analyses were done by P. R. All the authors discussed the results and wrote the manuscript.

**Funding information**   This work was supported by the Lithuanian Research Council, Grant No. S-MIP-20-36.

## A   Calculation of maximally localized Wannier functions

To optimize the Wannier functions centered at $x = na/2$ for small values of $\alpha$, we project the full Wannier state vector $|W_n(x)\rangle$ onto the following superposition of the first two ground states:

$$|\bar{n}\rangle = \frac{|1\rangle - (-1)^n |2\rangle \cos\alpha}{\sqrt{1 + \cos^2\alpha}} \ . \tag{40}$$

Such a superposition is featured in the second dark state close to the barriers at $x_{n\pm1} = (n \pm 1)\,a/2$ that confine the Wannier function in the region $(n-1)\,a/2 \lesssim x \lesssim (n+1)\,a/2$. For the lowest energy band, the optimal phase for every $q$-point in the 1BZ is then determined by optimizing the projected Wannier function $W_{\bar{n}}(x) = \langle\bar{n}|W_n(x)\rangle$ shown in Fig. 11 at the

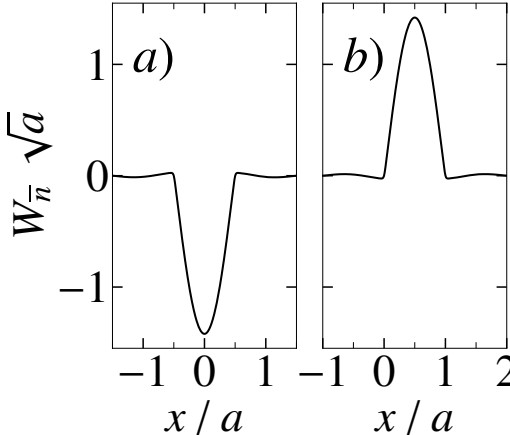

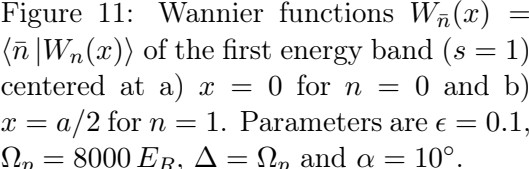

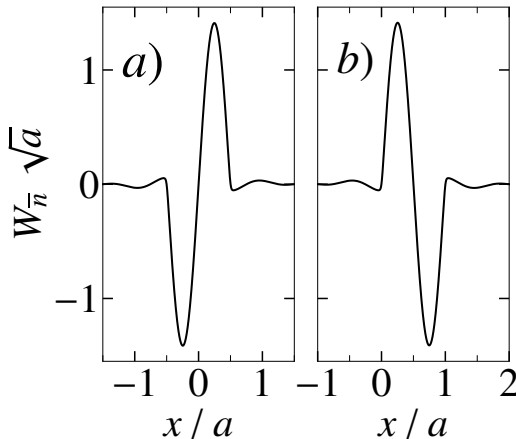

Figure 11: Wannier functions $W_{\bar{n}}(x) = \langle\bar{n}\,|W_n(x)\rangle$ of the first energy band ($s = 1$) centered at a) $x = 0$ for $n = 0$ and b) $x = a/2$ for $n = 1$. Parameters are $\epsilon = 0.1$, $\Omega_p = 8000\,E_R$, $\Delta = \Omega_p$ and $\alpha = 10°$.

Figure 12: Wannier functions $W_{\bar{n}}(x) = \langle\bar{n}\,|W_n(x)\rangle$ of the second energy band ($s = 2$) centered at a) $x = 0$ for $n = 0$ and b) $x = a/2$ for $n = 1$. Parameters are $\epsilon = 0.1$, $\Omega_p = 8000\,E_R$, $\Delta = \Omega_p$ and $\alpha = 10°$.

expected localization point $x = na/2$, giving

$$W_{\bar{n}}(na/2) = \frac{1}{\sqrt{N}}\sum_q e^{-iqna/2}\langle\bar{n}|\psi^{(q)}(na/2)\rangle = \frac{1}{\sqrt{N}}\sum_q \langle\bar{n}|g^{(q)}(na/2)\rangle\,, \tag{41}$$

where $N$ is the number of $q$ values in the extended Brillouin zone $-2\pi/a < q \leq 2\pi/a$, and $|g^{(q)}(x)\rangle$ is the periodic part of the Bloch state vector. Each term in the summation over $q$ is demanded to be real [45], which concludes the optimization procedure.

For the second energy band, the Wannier function $W_{\bar{n}}(x)$ vanishes at the center point $x = na/2$ representing a node (Fig. 12). As such, the optimal phase factors are determined by investigating the projected Wannier function at a point $x = na/2 \pm a/4$ shifted from the center $x = na/2$ by $a/4$ to the right or the left. This gives

$$W_{\bar{n}}(na/2 \pm a/4) = \frac{1}{\sqrt{N}}\sum_q e^{\pm iqa/4}\langle\bar{n}|g^{(q)}(na/2 \pm a/4)\rangle\,, \tag{42}$$

where, once again, all the components in the sum over $q$ are optimized by demanding them to be real. As $\alpha$ approaches $\pi/2$, one must optimize the Wannier function of the second band in a similar fashion, but now centered at $x = na/2 + a/4$ and without any additional shifting. In that case the Wannier function of the second energy band localizes between the neighboring potential barriers situated at $x = na/2$ and $x = (n+1)a/2$.

For small $\alpha$, the Wannier function $W_{\bar{n}}(x) = \langle\bar{n}\,|W_n(x)\rangle$ shown in Figs. 11–12 represents the dominant contribution to the full Wannier state vector $|W_n(x)\rangle$. Furthermore, the Wannier functions $W_{\bar{n}}(x)$ localized around the neighboring sites $n = 0$ and $n = 1$ are seen to have opposite signs due to the $\hat{T}_{a/2}$ symmetry of the Hamiltonian discussed in Sec. 2.2.

The methods described above work well for small or moderate values of $\alpha$. Figure 13 shows the Wannier functions $W_{\bar{n}}(x) = \langle\bar{n}\,|W_n(x)\rangle$ and $W_{\bar{n}^\perp}(x) = \langle\bar{n}^\perp\,|W_n(x)\rangle$ of the first

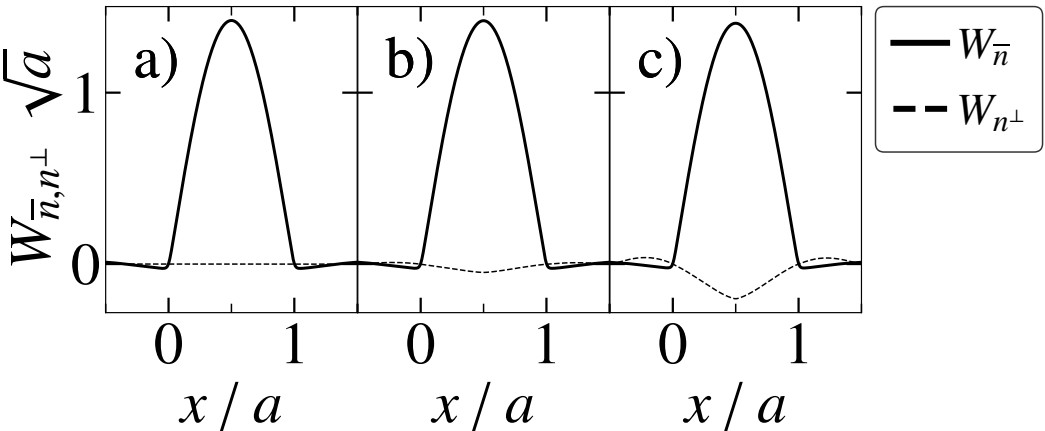

Figure 13: Wannier functions $W_{\bar{n}}(x)$ (solid lines) and $W_{\bar{n}^\perp}(x)$ (dashed lines) with $n = 1$ centered at $x = a/2$ for the first energy band. Parameters are $\epsilon = 0.1$, $\Omega_p = 8000\,E_R$, $\Delta = \Omega_p$ and a) $\alpha = 0$, b) $\alpha = 15°$, c) $\alpha = 30°$.

energy band for $\alpha = 0$, $\alpha = 15°$ and $\alpha = 30°$, where

$$|\bar{n}^\perp\rangle = \frac{|1\rangle \cos\alpha + (-1)^n |2\rangle}{\sqrt{1 + \cos^2\alpha}} \tag{43}$$

is a state vector orthogonal to $|\bar{n}\rangle$. One can see in Fig. 13 that the contribution from $W_{\bar{n}^\perp}(x)$ is tiny for $\alpha = 0$ and increases with $\alpha$, leading to the growth of the NN tunneling matrix element $J_1'$ shown in Fig. 9(a). This is because for small angle $\alpha$ the state vector $|\bar{n}^\perp\rangle$ almost completely overlaps with the state vector $|\bar{n'}\rangle$ at the neighboring sites $n' = n\pm1$, the latter $|\bar{n'}\rangle$ providing a dominant contribution to the neighboring Wannier state vectors $|W_{n\pm1}(x)\rangle$. An additional increase in the angle $\alpha$ leads to further growth of $W_{\bar{n}^\perp}(x)$ and subsequent spread of both $W_{\bar{n}}(x)$ and $W_{\bar{n}^\perp}(x)$. This enables tunneling between more distant sites, as illustrated in Fig. 9(a). When $\alpha$ approaches $90°$, the description of the odd energy bands in terms of the Wannier functions becomes problematic as the particle dynamics in the odd bands reduce to free atomic motion, as one can see in Fig. 8(d). On the other hand, for $\alpha \to 90°$ the Wannier functions of even bands become localized between the neighboring barriers separated by $a/2$, like in the $\Lambda$ scheme [5,6].

The Wannier functions of the dark, bright and excited states $W_{n,D_{1,2}}(x)$, $W_{n,B}(x)$ and $W_0(x)$ are displayed in Fig. 6 of the main text. They are found by projecting the full Wannier state vector $|W_n(x)\rangle$ (obtained using the above methods) to the corresponding atomic internal states $|D_l\rangle \equiv |D_l(x)\rangle$, $|B\rangle \equiv |B(x)\rangle$ and $|0\rangle$. The approximate Wannier functions of the dark states $W_{D_{1,2}}(x)$ calculated in terms of the adiabatic dark state Hamiltonian (18) are presented by the dotted lines in Fig. 6. They are optimized in the following way:

1. For the first band, we demand $W_{D_1}(x)$ to be real at $x = 0$.

2. For the second band and, we take the superposition $W_{D_1}(x) + W_{D_2}(x)$ at $x = a/4$ and demand it to be real.

The methods discussed here are implemented in the numerical codes that are available in the Supplementary Material [35].

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
