# Peer review of "Optical lattice with spin-dependent sub-wavelength barriers"

_SciPost Physics_

## Round 1 · Referee Report · Anonymous (Referee 1) · 2021-9-1

Strengths

  1. This paper proposes an interesting and original approach to realise tuneable spin-dependent sub-wavelength barriers, which can be relevant to current experiments.

  2. The theoretical analysis is well-referenced, thorough and includes a detailed discussion of various interesting different regimes, combining both exact calculations and illustrative models.

Weaknesses

This paper has no significant weaknesses.

Report

Overall, this is a nice paper, which extends previously studied schemes on sub-wavelength lattices to include spin-dependence. This is an interesting extension to the toolbox for cold atoms, and is worthy of publication. The changes requested are small typos and improvements to improve the manuscript as listed below.

Requested changes

  1. It would be helpful to show also the Wannier functions (e.g. as in Fig 4) for nonzero alpha to support the description currently given in words.

  2. It would be interesting to add more discussion in the conclusions of future prospects and what this technique could be useful for?

  3. Correct the typo in the sign of the first term in Eq. 7

  4. Around Eq. 14 and 13, several D_2 should be D_1 when referring to the first dark state or does the notation mean something different?

  5. Under Eq. 20, "of the angles and \phi" can just read "of the angles"?

  6. Refer to Figure 8 in the text between Eq. 40 and 41?

  • validity: high
  • significance: good
  • originality: good
  • clarity: high
  • formatting: excellent
  • grammar: excellent

Author:  Edvinas Gvozdiovas  on 2021-10-05  [id 1805]

(in reply to Report 1 on 2021-09-01)
Category:
remark
answer to question

Dear referee,

Thank you for reading our manuscript. We appreciate your helpful comments and suggestions.

In the revised manuscript we have added four new figures and two extra references. In the following we will refer to the new ordering of the figures and references.

Referee: 1. It would be helpful to show also the Wannier functions (e.g. as in Fig 4) for nonzero alpha to support the description currently given in words. Authors: We have added an extra Fig.13 in the Appendix showing the Wannier functions for non-zero alpha. In the paragraph following Eq. 43 we have discussed how does the alpha dependence of the Wannier functions shown in Fig.13 agree with other findings of the manuscript including the increase of the nearest neighbor tunneling elements with an increase of alpha.

Referee: 2. It would be interesting to add more discussion in the conclusions of future prospects and what this technique could be useful for? Authors: A discussion on the potential impact of the work on new possibilities for spin ordering and symmetry breaking has been added at the end of the Concluding Section of the manuscript. An extra sentence on this has also been added at the end of the Abstract.

Referee: 3. Correct the typo in the sign of the first term in Eq. 7. Authors: The typo in Eq. 7 has been fixed.

Referee: 4. Around Eq. 14 and 13, several D_2 should be D_1 when referring to the first dark state or does the notation mean something different? Authors: The dark state typos have been fixed there (see also our reply to the comments 6 and 7 of the Report 2).

Referee: 5. Under Eq. 20, "of the angles and \phi" can just read "of the angles"? Authors: This place did not look good, as the angle theta was missing. The sentence below Eq. 20 “angles and phi” was replaced by “angles theta and phi” in the revised manuscript.

Referee: 6. Refer to Figure 8 in the text between Eq. 40 and 41? Authors: Fig. 11 (the former fig.8) is now refered to just above Eq. 41.

---

## Round 1 · Referee Report · Anonymous (Referee 3) · 2021-9-14

Strengths

1) Innovative idea based on multilevel atoms 2) Detailed calculations of band structure and Wannier functions 3) Shared python scripts to reproduce and plot the data

Weaknesses

Sometimes a bit heavy to read as some variables are defined pages before and not recalled. It could be more easy to read.

Report

E. Gvozdiovas and coauthors study a new scheme to create sub-wavelength optical lattices based on the two dark states of a tripod atom. The two-dimensional dark sub-space provides an effective two-level system whose potential can be tuned by changing the power and the phase of the control and the two probe beams. Interestingly, by tuning the phase of one probe and the control beam standing wave, it is possible to change from a brick-wall synthetic lattice configuration, where the NNN tunneling is dominant over the NN one for small phase difference alpha. Conversely for alpha=pi/2 the effective potential change dramatically, providing a free dispersion for one dark state and NN tight binding lattice for the other dark state.

The manuscript is clear and well written: the authors give a thorough description of the model, calculate the band structure via exact diagonalization, calculate the Wannier wavefunctions and provide an extended set of python scripts to reproduce the calculation and analyze and plot the data, which I find great. I have only a few considerations, before the manuscript can be published:

1) The main concern I have is about the experimental feasibility of this scheme: the dressed bands shown in Fig. 3 have small gaps between them and are also fairly “flat” in terms of recoils. Since the authors mention a Sr87 experiment, what would be the temperature or the entropy needed to load the ground state of the band structure and observe the effect of these spin-dependent lattices?

2) a is not defined in eq. 3,4,5. Although it is clear from the context that is the lattice spacing, I would add it for clarity.

3) In equation 23 and 24 it should be recalled that epsilon is Omega_p/Omega_c. It is defined in Eq. (5) and by the time the reader gets to Eq. 23 it can be forgotten. Also it is not clear how eq. 24 follows from eq. 23. Maybe the authors can guide the reader or give an intuition for how to derive eq. 24.

4) What is the difference between Fig. 3a and 3b? They look the same and show 4 bands but the caption says first 3 bands. It should be explained better.

5) It would be very helpful to have a plot of the effective potential in real space to appreciate the unusual shape with large wells and narrow barriers.

6) I assume to observe the effect of this sub-wavelength potentials, the atoms need to be cooled down. I wonder if a tripod EIT cooling scheme recently observed in trapped ions (https://journals.aps.org/prl/abstract/10.1103/PhysRevLett.125.053001) might turn out to be helpful also to cool tripod neutral atoms in these spin dependent optical lattices, provided that the lattices are deep enough.

  • validity: top
  • significance: high
  • originality: high
  • clarity: high
  • formatting: good
  • grammar: excellent

Author:  Edvinas Gvozdiovas  on 2021-10-05  [id 1803]

(in reply to Report 3 on 2021-09-14)
Category:
remark
answer to question

Dear referee,

Thank you for reading our manuscript. We are very thankful for your insights and suggestions.

In the revised manuscript we have added four new figures and two extra references. In the following we will refer to the new ordering of the figures and references.

Referee: 1) The main concern I have is about the experimental feasibility of this scheme: the dressed bands shown in Fig. 3 have small gaps between them and are also fairly “flat” in terms of recoils. Since the authors mention a Sr87 experiment, what would be the temperature or the entropy needed to load the ground state of the band structure and observe the effect of these spin-dependent lattices? Authors: The subwavelength barriers for the Lambda scheme have already been implemented by adiabatically loading of the ultracold atoms to the lowest band [19]. A similar adiabatic loading can be applied for the present situation involving the tripod scheme. The difference in energies between the ground and first excited states is defined by the recoil energy, which is of the order of tens of kHZ for an atom like Rb87 and corresponds to the microkelvin temperature range. This greatly exceeds the temperature of the ultracold atomic gas which is of the nanokelvin range. Therefore the ultracold atoms adiabatically loaded to the lowest Bloch band will be maintained in it. Regarding the entropy: since the tripod scheme has two dark states, there is an extra possibility to populate specific dark states during the adiabatic loading. This can be done by transferring the atoms to a desired dark state by the tripod STIRAP (stimulated adiabatic Raman passage) prior to the adiabatic loading [23,24]. We have added a paragraph containing such a discussion in the final part of the Concluding Section 5, see “The lattice with sub-wavelength barriers ...“

Referee: 2) a is not defined in eq. 3,4,5. Although it is clear from the context that is the lattice spacing, I would add it for clarity. Authors: Lattice spacing is now defined below equations 3, 4, 5.

Referee: 3) In equation 23 and 24 it should be recalled that epsilon is Omega_p/Omega_c. It is defined in Eq. 5 and by the time the reader gets to Eq. 23 it can be forgotten. Also it is not clear how eq. 24 follows from eq. 23. Maybe the authors can guide the reader or give an intuition for how to derive eq. 24. Authors: epsilon is now expressed explicitly below Eq. 24 and in some other places of the manuscript. Also, above Eq. 24 we have added a short explanation how to derive this equation.

Referee: 4) What is the difference between Fig. 3a and 3b? They look the same and show 4 bands but the caption says first 3 bands. It should be explained better. Authors: The purpose of parts a) and b) of Fig.4 (the former Fig. 3) is to display the overall band structure showing that it does not change rapidly with the phase alpha, while in c) and d) the enlarged first band exhibits an apparent alpha dependent fine structure. Dispersions displayed in Fig. 4 correspond to small values of the alpha (0 and 15 degrees), so they indeed look quite similar on the large scale in parts (a)-(b). The difference in the dispersion curves becomes more visible on the large scale for bigger values of alpha shown in Fig 5 and Fig. 8. To explain better this point we have added an extra sentence and a footnote at the beginning of the paragraph after Eq.(36) followed by the sentence: „Figures 4(c)-(d) display the enlarged dispersion of the first energy band s = 1“. Figures 4(a)-(b) do show 3 bands rather than 4 bands. The origin of the confusion might be the absence of a horizontal gap separating the parts (a)-(b) from the parts (c)-(d) of Fig. 4 (the former Fig.3) in the original manuscript. In the revised manuscript we have introduced such a horizontal gap to separate different plots.

Referee: 5) It would be very helpful to have a plot of the effective potential in real space to appreciate the unusual shape with large wells and narrow barriers. Authors: In Sec. 3.2.2 we have added a new figure (Fig.3) showing the real space plots for three different values of alpha of the Rabi frequencies Ω_{1,2,3} (x), as well as the quantities defining the geometric scalar and vector potentials in Eqs.(20) and (19). This should help to appreciate the shape of the scalar and vector potentials.

Referee: 6) I assume to observe the effect of this sub-wavelength potentials, the atoms need to be cooled down. I wonder if a tripod EIT cooling scheme recently observed in trapped ions (https://journals.aps.org/prl/abstract/10.1103/PhysRevLett.125.053001) might turn out to be helpful also to cool tripod neutral atoms in these spin dependent optical lattices, provided that the lattices are deep enough. Authors: In this paper for trapped ions [34] and the previous work for electrically neutral atoms [33] the tripod EIT cooling was carried out using the laser beams with spatially homogenous laser beams, whereas in the present scheme we are dealing with the standing waves. It is hard to say without making a more detailed analysis if the current scheme could provide an efficient cooling. On the other hand, additional cooling is not necessarily needed for the present purpose. In fact, as discussed when replying to the Comment (1), the ultracold atoms can be adiabatically loaded to the lowest band of the lattice. In the Revised manuscript we have mentioned the previous work on the cooling of atoms and ions [33,34], when talking about the tripod scheme in the introductory part.

---

## Round 1 · Referee Report · Anonymous (Referee 2) · 2021-9-14

Strengths

  1. The paper describes a topic in the frontier of Physics with relevance to a broad scientific audience.

  2. The paper is relevant to Physics no only in theory but also in experiment.

Weaknesses

  1. The paper could be a bit more specific on the sections where the reader will find a result.

  2. The paper could be a bit more specific on the equations from which a result is being displayed.

  3. The order of some figures in the paper could improve to match the order of ideas in the text.

Report

The article proposes and characterizes a method to implement a periodic optical potential for ultracold neutra atoms with a lattice constant below the diffraction limit. Such structure is created by a light field that couples a tripod system, that is restricted to a superposition of dark states. While one of the states is barely affected by the periodic landscape, the other experiences the optically engineered sub-wavelength periodic potential.

The present work describes a versatile, novel and realist technique that has a potential impact on different research areas ranging from: high-energy physics, to quantum information science. This proposal would naturally give rise to experimental follow-up work on the quantum simulation front.

The paper is clearly written, with few adjustments required to eliminate minor ambiguities. The introduction is complete, clear and informative of the state of the art of the field of research.

The abstract could add a sentence or two to emphasize on the potential impact of the work on other fields of physics.

All along the manuscript the authors cite relevant literature on previous achievements both in the field of theoretical and experimental physics as far as I am aware.

The authors include sufficient detail along the manuscript to follow their arguments. 

The conclusion succinctly summarizes the work. It includes references to relevant experimental work based on the tripod scheme which support the realism of the proposal, while it describes the opportunities and perspectives to implement the present sub-wavelength lattice.

After minor adjustments in the manuscript, I would recommend publication in SciPost Phys.

Requested changes

  1. Below eqn. (6) it is not clear to me why “The zeros of the control field correspond to theta = 0 and theta = pi.” I would have expected it to be at theta = pi/2.

  2. In the paragraph below eqn. (5), where it reads: “This results in a periodic array of state-dependent sub-wavelength barriers at x=…, as will be shown later.” Please write down the specific number of the section where this will be shown.

  3. Below eqn. (7) please explicitly define m.

  4. Just before the beginning of Section 3, where it says “As we will see later, such a condition is well maintained for atomic dynamics in the dark state manifold in which atomic decay is suppressed.” Please write down the specific number of the section where this result will be developed.

  5. Below eqn. (13), where it says "They are orthogonal between each other” it must be <D1|D2>=0.

  6. In the next paragraph starting with “The dark states are not uniquely defined”, state |D1> is mislabelled 3 times. Please correct.

  7. I believe state |D1> is defined by eqn. (14). Please correct.

  8. Up to eqn. (16), the dependence of |D_l> on x has not being explicit. I suggest to keep it that way for the coherence of the notation.

  9. In the paragraph above eqn. (28), please define Q in expression of E.

  10. Below eqn. (37), it must say Jn (no prime) = Jn’ + iJn’’. Please correct.

  11. In the same paragraph (below eqn.(37), please say a few words on the sign of J1. You need J1<0 and J2>0 to get the dispersion of Fig. 3b, correct?

  12. In the paragraph above section 4.2 where it says: “The solid dispersion lines in Fig. 3 fit well to the dispersions obtained for the adiabatic motion within the dark state manifold.” I was expecting to find both models in Figure 3. Why are these adiabatic motion dispersions not displayed in Fig. 3?

  13. Again, in the paragraph above section 4.2. The figure numbering goes from Figure 3 to Figure 6. I suggest a reordering of the figures to prevent this jump in the text.

  14. In the caption of Figure 4, it says: “The Wannier functions of the dark states shown in a) and c) agree well with the ones obtained by diagonalizing the effective dark state Hamiltonian”(18). I was expecting a direct comparison between such Wannier functions. Can the exact Wannier functions be added to compare?

  15. In the caption of Figure 6. Can the text be more specific as to clarify to which Hamiltonian do the exactly calculated Bloch bands correspond?

  16. In the second paragraph before section 5. It says that the 4x4 eigenvalue equation reproduces the band structure accurately …Can the band structure calculated that way be shown in Fig. 6 to compare directly?

  • validity: high
  • significance: high
  • originality: high
  • clarity: high
  • formatting: good
  • grammar: excellent

Author:  Edvinas Gvozdiovas  on 2021-10-05  [id 1804]

(in reply to Report 2 on 2021-09-14)
Category:
remark
answer to question

Dear referee,

Thank you for reading our manuscript. We appreciate your helpful comments and suggestions.

In the revised manuscript we have added four new figures and two extra references. In the following we will refer to the new ordering of the figures and references.

Referee: 1. Below eqn. (6) it is not clear to me why “The zeros of the control field correspond to theta = 0 and theta = pi.” I would have expected it to be at theta = pi/2. Authors: Indeed there was an error / misprint there, as pointed out by the Referee. In the revised version we have replaced the statement below Eq. 6 by: “The zeros of the control field correspond to theta = pi/2”.

Referee: 2. In the paragraph below eqn. (5), where it reads: “This results in a periodic array of state-dependent sub-wavelength barriers at x=…, as will be shown later.” Please write down the specific number of the section where this will be shown. Authors: Sec. 3.2.2 is now referred to below Eq. 5.

Referee: 3. Below eqn. (7) please explicitly define m. Authors: The atomic mass is defined below Eq. 7.

Referee: 4. Just before the beginning of Section 3, where it says “As we will see later, such a condition is well maintained for atomic dynamics in the dark state manifold in which atomic decay is suppressed.” Please write down the specific number of the section where this result will be developed. Authors: Just before Section 3 we now refer to Section 4 and the corresponding figures 4 and 8 contained in this Section.

Referee: 5. Below eqn. (13), where it says "They are orthogonal between each other” it must be <D1|D2>=0. Authors: Our apologies for another misprint. The relation <D1|D2>=0 has been fixed.

Referee: 6. In the next paragraph starting with “The dark states are not uniquely defined”, state |D1> is mislabelled 3 times. Please correct. Authors: We apologize for the sloppy typos, they have been fixed.

Referee: 7. I believe state |D1> is defined by eqn. (14). Please correct. Authors: Equation 14 was corrected.

Referee: 8. Up to eqn. (16), the dependence of |D_l> on x has not being explicit. I suggest to keep it that way for the coherence of the notation. Authors: In Eq. 16 and most of the subsequent relations, D_{l}(x) was changed by D_{l} to stay consistent in notation. The variable x is explicitly displayed in the dark states only when one needs to emphasize their position dependence, such as in the relation below Eq. 20.

Referee: 9. In the paragraph above eqn. (28), please define Q in expression of E. Authors: Q is now defined below Eq. 28.

Referee: 10. Below eqn. (37), it must say Jn (no prime) = Jn’ + iJn’’. Please correct. Authors: Typo below Eq. 37 for J_n has been fixed.

Referee: 11. In the same paragraph (below eqn.(37), please say a few words on the sign of J1. You need J1<0 and J2>0 to get the dispersion of Fig. 3b, correct? Authors: The Referee is correct concerning the sign of the tunnelling parameters, as is shown in Figs. 9 and 10. We have included an additional discussion just below Eq. 37 addressing this question.

Referee: 12. In the paragraph above section 4.2 where it says: “The solid dispersion lines in Fig. 3 fit well to the dispersions obtained for the adiabatic motion within the dark state manifold.” I was expecting to find both models in Figure 3. Why are these adiabatic motion dispersions not displayed in Fig. 3? Authors: We have added dotted lines to Fig. 4 (former Fig.3). Also, we have included an extra sentence just before the last paragraph of Sec. 4.1: “Note that in Fig. 4, deviation of the adiabatic dispersion curves (dashed lines) from the exact dispersion (solid lines) are mainly due to non-zero values of Delta and Gamma, the difference becoming minimum for zero detuning Delta = 0 shown in Fig. 5.”

Referee: 13. Again, in the paragraph above section 4.2. The figure numbering goes from Figure 3 to Figure 6. I suggest a reordering of the figures to prevent this jump in the text. Authors: We understand the problem. On the other hand, the logic of the paper is to consider first small and moderate values of the parameter alpha in Sec. 4.1 following by the analysis of the Wannier functions (Sec.4.2) which is valid in this regime, finally going to the larger values of alpha in Sec. 4.3. At the end of Sec.4.1 we wish to mention in advance some of the figures of Sec. 4.3 showing the connection between the two regimes. To make the reading smoother, in the revised manuscript we have made a reference to Sec.4.3 in addition to mentioning the figure 8 (the former figure 6): “More generally, we have checked that the effective dark state Hamiltonian (18) recreates the exact band structure for any α, not necessarily small, as shown in Fig. 8 of Sec. 4.3, …”

Referee: 14. In the caption of Figure 4, it says: “The Wannier functions of the dark states shown in a) and c) agree well with the ones obtained by diagonalizing the effective dark state Hamiltonian”(18). I was expecting a direct comparison between such Wannier functions. Can the exact Wannier functions be added to compare? Authors: In Fig 6 (the former Fig. 4) we have added dotted lines representing Wannier functions obtained by diagonalizing the effective dark state Hamiltonian. However, it is hard to see these lines since they coincide almost perfectly with the exact Wannier functions for the dark states. Additionally, we have added vertical grid lines in this figure to improve readability.

Referee: 15. In the caption of Figure 6. Can the text be more specific as to clarify to which Hamiltonian do the exactly calculated Bloch bands correspond? Authors: In the caption of Fig. 8 (the former Fig. 6) we made a reference to the Hamiltonian (7): “The first six Bloch bands (s=1,2,…), obtained by diagonalizing the full Hamiltonian (7) …”

Referee: 16. In the second paragraph before section 5. It says that the 4x4 eigenvalue equation reproduces the band structure accurately …Can the band structure calculated that way be shown in Fig. 6 to compare directly? Authors: We think Fig. 8 (former Fig. 6) would be too dense if adding also the solutions of the 4x4 eigenvalue equation. Therefore to address this issue we have added an extra figure (Fig. 5) at the end of Sec. 4.1. It shows the energy dispersions obtained using all 3 methods (exact diagonalization, using adiabatic dark state Hamiltonian and the 4x4 eigenvalue equation) for zero detuning (Delta=0).

*) Additional comment in the Report 2 Referee: The abstract could add a sentence or two to emphasize on the potential impact of the work on other fields of physics. Authors: An extra sentence has been added at the end of the Abstract (on the potential impact of the work on new possibilities for spin ordering and symmetry breaking). A more extended discussion on this issue has been placed at the end of the Concluding Section of the manuscript.

Anonymous on 2021-10-27  [id 1883]

(in reply to Edvinas Gvozdiovas on 2021-10-05 [id 1804])
Category:
remark

To the authors,

Thank you for considering and applying the requested changes.
After this update in the manuscript I stand by my deliberation and recommend the present work "Optical lattice with spin-dependent sub-wavelength barriers" to be published in SciPost.

---

## Round 2 · Author Response

We would like to thank the Referees for carefully reading our manuscript and providing helpful comments and suggestions. We have revised the manuscript taking into account the Referee comments. In the revised manuscript we have added four new figures and two extra references. We refer to the new ordering of figures and references in the list of changes and remarks for the referees.

---

## Round 2 · List of Changes

MAIN CHANGES:

1. Two new references [33,34] on cooling atoms and ions using the tripod scheme have been added in the Introduction.

2. A new figure in Sec. 3.2.2 shows the functions characterizing the effective potentials (Figure 3). We have also added references to this figure in the appropriate places of Sec. 3.2.2.

3. In the numerical code, we use the wavenumber 2pi/a, while in the paper we use the wavenumber pi/a to define the recoil energy. As such, some of the system parameters written under the Figures were originally missing a factor of 4. We have corrected the parameters under the figure captions and have redrawn the figures with the correct parameters where necessary to address this issue. As a consequence, Figs. 4 through 12 have been changed in the way just described.

4. A paragraph has been added before Section 4 to discuss the limit for alpha -> pi/2.

5. A new figure before Sec. 4.2 (Figure 5) shows the energy dispersions for zero detuning, obtained by the 3 different methods presented in the paper. Additionally, the paragraph from Section 4.3 has been moved next to Figure 5. This paragraph has also been rewritten to represent more accurately our results, namely we emphasize that the 4x4 scattering approach recreates very accurately the exact odd band structure for large values of the phase parameter. This is emphasized again in Sec. 4.3 where the tight binding tunnelling parameters are discussed.

6. A new figure in Sec 4.3 (Figure 9) shows the tight binding tunnelling parameters of the first two energy bands as a function of phase, demonstrating the free particle-like behaviour of the odd bands, and the even bands approaching Lambda-like bands with only non-zero nearest neighbor (NN) tunnelling for alpha -> pi/2. Two paragraphs were added in the same Section to discuss these results.

7. The last paragraph in Section 4.3 about the detuning has been extended to explain its effects on the Wannier functions and the tunnelling parameters. We have also made a connection there with the Lambda-like setup.

8. The “concluding remarks” section has been extended to discuss the adiabatic loading of ultracold atoms and future prospects.

9. At the end of the Concluding Remarks section a note has been added on a recent preprint by Kubala et al.[44].

10. Appendix A has been altered considerably. First, a state vector orthogonal to that given by Eq. 40 has been introduced in Eq. 43. The corresponding two components of the Wannier function have been plotted for the first energy band in Fig. 13 for various of alpha, demonstrating the effects of non-zero alpha on the Wannier functions, and helping understanding the connection between the Wannier functions and the tunnelling parameters. Secondly, we have further explained how the Wannier functions from the adiabatic dark state Hamiltonian are optimized.

MINOR CHANGES:
1. A missing factor of 1/2 was added in the second part of Eq. 1.
2. There were notational inconsistencies between Eqs. 17 and 38. Namely, Eq. 38 was previously marked with a ket, while Eq. 17 was not. This issue has been fixed.
3. A typo in “the expanded 1BZ covering the range $2\pi/a<q\le2\pi/a$” was fixed, which now reads “the expanded 1BZ covering the range $-2\pi/a<q\le2\pi/a$”
4. Figure 6 now contains dotted lines that represent the Wannier functions obtained from the adiabatic dark state Hamiltonian. We have also added grid lines for better readability.
5. Figure 4 now contains dotted lines that represent solutions of the adiabatic dark state Hamiltonian for non-zero detuning.
6. Epsilon has been replaced with epsilon=Omega_p / Omega_c in many places in the paper to help the reader recollect the definition.
7. Equation 20 was rewritten in terms of newly defined row and column vectors constituting the geometric scalar potential.
8. Figure 7 caption was shortened and the discussion contained in this caption was moved to the main text in Section 4.3.
9. An acknowledgement to Ian B. Spielman was added.

---

## Editorial Decision

publication_decision_taken:_accept